# Screening for Patients with Visual Acuity Loss in Primary Health Care: A Cross Sectional Study in a Deprived Hungarian Population

**DOI:** 10.3390/healthcare11131941

**Published:** 2023-07-05

**Authors:** Rahul Naresh Wasnik, Veronika Győri-Dani, Ferenc Vincze, Magor Papp, Anita Pálinkás, János Sándor

**Affiliations:** 1Department of Public Health and Epidemiology, Faculty of Medicine, University of Debrecen, 4032 Debrecen, Hungary; vincze.ferenc@med.unideb.hu; 2Doctoral School of Health Sciences, University of Debrecen, 4032 Debrecen, Hungary; 3St. Francis Hospital, 1089 Budapest, Hungary; gyori.veronika@maltai.hu; 4Semmelweis Health Promotion Center, Semmelweis University, 1089 Budapest, Hungary; papp.magor@semmelweis-univ.hu; 5ELKH-DE Public Health Research Group, Department of Public Health and Epidemiology, Faculty of Medicine, University of Debrecen, 4032 Debrecen, Hungary; palinkas.anita@med.unideb.hu

**Keywords:** visual acuity loss, screening, primary care, general health check, deprivation

## Abstract

Screening for visual acuity loss (VAL) is not applied systematically because of uncertain recommendations based on observations from affordable countries. Our study aimed to evaluate the effectiveness of primary health care-based screening. A cross-sectional investigation was carried out among adults who did not wear glasses and did not visit an ophthalmologist in a year (N = 2070). The risk factor role of sociodemographic factors and the cardiometabolic status for hidden VAL was determined by multivariable linear regression models. The prevalence of unknown VAL of at least 0.5 was 3.7% and 9.1% in adults and in the above-65 population. Female sex (b = 1.27, 95% CI: 0.35; 2.18), age (b = 0.15, 0.12; 0.19), and Roma ethnicity (b = 2.60, 95% CI: 1.22; 3.97) were significant risk factors. Higher than primary school (b_secondaryschoolwithoutgraduation_ = −2.06, 95% CI: −3.64; −0.47; and b_secondaryschoolwithgraduation_ = −2.08, 95% CI: −3.65; −0.51), employment (b = −1.33, 95% CI: −2.25; 0.40), and properly treated diabetes mellitus (b = −2.84, 95% CI: −5.08; −0.60) were protective factors. Above 65 years, female sex (b = 3.85, 95% CI: 0.50; 7.20), age (b = 0.39, 95% CI: 0.10; 0.67), Roma ethnicity (b = 24.79, 95% CI: 13.83; 35.76), and untreated diabetes (b = 7.30, 95% CI: 1.29; 13.31) were associated with VAL. Considering the huge differences between the health care and the population’s social status of the recommendation-establishing countries and Hungary which represent non-high-income countries, the uncertain recommendation of VAL screening should not discourage general practitioners from organizing population-based screening for VAL in non-affordable populations.

## 1. Introduction

According to the World Health Organization (WHO), 2.2 billion people globally suffer from visual impairment mainly caused by uncorrected refractive errors, cataracts, and age-related macular degeneration (AMD). The affected population is dominated by elderly people. This unavoidable cause is becoming a more dominant determinant as the world population ages [1].

Conversely, investigations conducted at both country and within-country social group levels conclusively show that social deprivation is a significant risk factor for visual impairment [2,3]. Social inequality used to be attributed to population-level variability in health literacy, the availability of health care, and the organization of ophthalmologic care. Although the specific mechanisms behind this association have not been properly explored, primary health care plays an important role in early detection, diagnosis, and screening, which are the critical components for an optimal approach to visual impairment [4,5]. To enable eye care service providers, general medical practitioners, and public health authorities to reduce this inequality, the WHO recommends regular monitoring of the effectiveness of ophthalmologic care. This recommendation is a part of the global plan of action for universal eye health and emphasizes the importance of providing basic eye care at affordable prices for all communities [6,7,8,9].

Progressive vision loss affects many facets of daily life, such as reading, mobility, and independence, greatly impacting quality of life, including economic status and education level, leading to the loss of productivity [10,11]. Progressive vision loss impairs mental health and increases the risk of traumas as well [12,13,14,15].

Considering that the prevalence of undiagnosed visual impairment is very high, and visual acuity checking is safe and cheap, population-based screening seems to be an important tool to reduce the consequences of undiagnosed visual impairment [16].

Direct evidence of the effectiveness of screening is missing because several randomized controlled trials (on outcomes of visual acuity, morbidity, mortality, general or vision-related quality of life, functional status, and cognition) failed to demonstrate the benefit of programmatic vision screening in adults. These trials were essentially from resource-rich settings of affordable countries and were generally limited by small sample sizes, low intervention uptake, substantial losses to follow-up, and the use of self-reported primary outcomes [17]. The causal association between uncorrected visual impairment, mental and behavioral changes, and elevated mortality rates is still subject to uncertainty. In fact, there is a dearth of circumstantial data supporting the advantages of using refractive lenses [18,19,20,21,22,23,24].

The US Preventive Services Task Force (USPSTF) and other organizations, such as the American Academy of Family Physicians and the Canadian Task Force on Preventive HealthCare, for population-based screening recommendations acknowledge these ambiguities. Their statement advocates that the existing evidence is insufficient to assess the balance of benefits and harms of screening for impaired visual acuity in asymptomatic adults 65 years or older [25,26,27]. Perhaps other organizations such as the American Academy of Ophthalmology and the Royal Australian college of General Practitioners potentially hold a more favorable stance towards regular screening, emphasizing the effectiveness of refractive lenses, cataract surgery, and treatment for AMD [28,29].

Considering the well-known social inequalities of visual acuity loss and the huge impact of care accessibility on the prognosis of eye disorders, the effectiveness of screening for visual acuity loss is highly dependent on the social and institutional status of a country. It is explicitly acknowledged by the USPSTF that well-designed primary care studies are required to clarify the potential benefits of screening that evaluate new vision screening accompanied by proper referral to appropriate follow-up care and that are targeted to higher risk populations [30].

In Hungary, visual acuity examination organized by population-based screening is recommended in general practice by a ministerial decree (Annex No: 2 of Act 51/1997 (XII.18) NM). The first examination is scheduled for 21 years of age, which should be followed by a yearly examination for those beyond the age of 65. Unfortunately, there is no monitoring for the implementation of decree-defined screening, and there are no routine statistics on the prevalence of visual acuity loss either in the general population or in the sociodemographic strata.

According to a recent representative survey among 50+ year-old Hungarian adults, the prevalence of less than 0.5 visual acuity is 13.4%. The leading causes were cataracts (47.1%), refractive error (30.2%), AMD and another posterior segment disease (14.5%), followed by glaucoma (2.7%), and diabetic retinopathy (2.4%). This etiological background is close to the reported western European pattern [31].

The prevalence of less than 0.5 visual acuity was 8.8% among 20- to 64-year-old Hungarian adults in a regional survey. The frequency of wearing glasses among participants with a visual acuity of <0.5 was 77.1% but was much less (14.3%) among participants from the ethnic minority (Roma) population, which demonstrated the presence of a huge social inequality in Hungarian eye care [32].

The ineffectiveness of Hungarian eye care in the early diagnosis of visual impairment is reflected in the observed much higher prevalence of vision impairment due to uncorrected presbyopia (when the best-corrected-distance visual acuity is 6/12 or better), which is not accompanied by an observed much higher prevalence of blindness (<3/60 vision in the better eye in presentations) and mild-to-severe vision loss (<6/18–3/60 vision in bilateral presentations) compared to resource-rich settings, where relevant RCTs have been implemented [31].

The aim of our study on deprived, adult populations was (1) to determine the prevalence of visual acuity loss, (2) to identify the influencing factors of visual acuity loss, and (3) to evaluate the effectiveness of primary health care-based visual acuity loss screening.

## 2. Materials and Methods

### 2.1. Setting

Our study was conducted as part of the Public Health Focused Model Program for Organizing Primary Care Services Backed by a Virtual Care Service Center. The program established 4 general medical practice clusters (GPCs) in the most disadvantaged region of Hungary populated by 32,655 adults. The aim of the model program was to improve the health status of the deprived population by integrating prevention and health promotion into primary health care [33].

One component of the program’s services was a population-based, organized general health check (GHC) for all adults, regardless of their health status [34]. As part of the GHC, a questionnaire was applied. A physical examination was performed; medications along with laboratory parameters and cardiometabolic primary care data were collected. GHC was launched on 1 October 2013, and 5036 patients participated by 30 September 2014 [35].

The present investigation is a secondary analysis of the database built by the GHC.

### 2.2. Data Collection

The GHC was carried out by trained public health practitioners and nurses [36]. The demographic status was described by the age, sex, and self-declared ethnicity of patients (distinguishing Roma and non-Roma). The socioeconomic status of participants was assessed by employment status (employed and unemployed) and level of education (noncompleted primary school, completed primary school, secondary school without graduation, secondary school with graduation, higher education). Because the regular ophthalmologic examination is a compulsory part of the care for high-prevalence chronic diseases (hypertension and diabetes mellitus), patients’ blood pressure and fasting blood glucose concentration were measured, and the cardiometabolic history was registered in the GHC, and they were categorized into one of the following subgroups: (a) no existing disease or treatment (normotensive, normoglycemic); (b) diagnosed and properly treated; (c) diagnosed but inadequately treated; or (d) an unknown and therefore untreated disease group. A blood pressure of 140/90 mmHg (average of 2 measurements) and a fasting blood glucose of 7 mmol/L were set as cutoff values for normal values. Patients without diagnosed diseases whose readings were below the cutoff points were classified as normotensive or normoglycemic.

### 2.3. Visual Acuity Assessment

Visual acuity was measured with a Snellen-type Kettesy table consisting of optotypes according to the standard ophthalmologic practice. (Each row of these test charts is designed from larger optotypes into smaller optotypes to evaluate patients’ reading and observing abilities.) Visual acuity (VA) was expressed as the ratio of the distance from the patient to the test chart and the distance at which the smallest optotype that the patient can read can be detected for a healthy eye. According to the Council of the European Communities Directive on driving licenses (Directive 91/439/EEC), a person is eligible for a driving license if his or her best-corrected binocular visual acuity reaches at least 0.5. The same threshold of visual acuity loss is applied in occupational medical practice for calculating the degree of disability of disadvantaged workers in Hungary.

All participants were characterized with a continuous outcome variable as visual acuity loss (cVAL, assessing the severity of total ocular damage by calculating 1-VA) and with a dichotomous outcome variable as severe (cVAL < 0.5) visual impairment (dSVI).

### 2.4. Data Analysis

The visual acuity investigation was performed on patients who did not wear glasses or contact lenses and were not known to have an eye care provider. Therefore, all patients who wore glasses or contact lenses or had seen an ophthalmologist within one year before the examination date were excluded from the study. For the analyses, we used data from patients with complete records.

Adjusted linear regression coefficients (b) with the corresponding 95% confidence intervals (95% CI) from multivariate linear regression analysis were used to identify the protective and risk factors for cVAL. Factors influencing the dSVI were determined by multivariate logistic regression analysis. The results are presented as odds ratios (ORs) with corresponding 95% CIs.

Each statistical analysis was performed on the entire study population and on the subgroup of patients older than 65 years.

Statistical analyses were performed using PASW Statistics (version 18.0, SPSS Inc., Chicago, IL, USA).

### 2.5. Ethical Approval

The study protocol was approved by the Ethics Committee of the Hungarian National Scientific Council for Health (TUKEB 16676-3/2016/EKU, 0361/16). All participants signed an informed consent form before the health check data collection started.

## 3. Results

The sampling process is summarized in Figure 1. After the exclusion of patients with incomplete records, 2070 patients remained, who met all of the inclusion criteria and served as the target population for acuity loss screening.

The descriptive statistics (for the whole target group and the subgroup of patients over 65 years of age) are summarized in Table 1. (More detailed descriptive statistics are presented in Appendix A available in Appendix A). A visual acuity of less than 0.5 could be detected in 3.7% of the target group and in 9.1% of adults over 65 years of age.

The distribution of the population targeted by screening and the yield of screening across demographic strata are summarized in Figure 2. The majority of screening-detected dSVIs (59.2%) were observed among persons less than 65 years old.

In multivariate linear regression analyses, female sex (b = 1.27, 95% CI: 0.35; 2.18), age (b = 0.15, 95% CI: 0.12; 0.19), and Roma ethnicity (b = 2.60, 95% CI: 1.22; 3.97) were found to be significant risk factors for cVAL. Higher than primary school (b_secondary school without graduation_ = −2.06, 95% CI: −3.64; −0.47; and b_secondary school with graduation_ = −2.08, 95% CI: −3.65; −0.51) and employment (b: −1.33, 95% CI: −2.25; −0.40) were significant protective factors. Properly treated diabetes mellitus (b = −2.84, 95% CI: −5.08; −0.60) also proved protective in the multivariate analysis (Table 2).

In the age group above 65 years of age, female sex (b = 3.85, 95% CI: 0.50; 7.20), age (b = 0.39, 95% CI: 0.10; 0.67), Roma ethnicity (b = 24.79, 95% CI: 13.83; 35.76), and untreated diabetes (b = 7.30, 95% CI: 1.29; 13.31) were associated with cVAL (Table 2).

Examining the entire target group using multivariate logistic regression analysis, we found that secondary-level education compared to primary-level education was a significant protective factor against dSVI (OR = 0.35, 95% CI: 0.13; 0.93. However, age (OR = 1.06, 95% CI: 1.04; 1.06) and Roma ethnicity (OR = 2.80, 95% CI: 1.41; 5.55) were significant risk factors (Table 3).

In the subgroup of patients older than 65 years of age, age (OR = 1.10, 95% CI: 1.03; 1.18) and Roma ethnicity (OR = 7.81, 95% CI: 1.36; 44.68) were found to be significant risk factors for dSVI (Table 3). (Results from the regression models for adults not above 65 years of age are presented in Appendix A available in Appendix A).

## 4. Discussion

### 4.1. Proportion of the Target Screening Group in the Whole Population

According to our investigation, 53.6% (95% CI: 52.0–55.1) of the adults (for males 58.3%, 95% CI: 55.7–61.0; for females 51.1% 95% CI: 49.2–53.0) and 30.2% (95% CI: 27.5–32.9) of the 65+ adults (for males 35.9%, 95% CI: 31.1–41.7; for females 27.2% 95% CI: 24.0–30.4) did not wear glasses or consult an ophthalmologist in a year.

To evaluate the observed proportions, the prevalence of using glasses or consulting an ophthalmologist was computed for other age groups. A former Hungarian survey reported a 24.0% prevalence for the 20–64-year age group [32], which was lower, while another survey published 84.3% for the 50+-year age group [31], which was higher than the corresponding values observed in our investigation (36.9% and 59.8%, respectively).

In the Abruzzo region of Italy, the prevalence of known visual problems among individuals aged above 50 (44%) is less than the finding in our research prevalence (59.8%) [37]. In the UK, 88% of people aged 75 or older owned glasses, which corresponds to the similar prevalence in our sample (76.0%) [38]. The prevalence of the use of glasses is 92.4% in the USA among Medicare beneficiaries above 65 years of age, which is above our observation (69.8%) [39]. Our observation (53.5%) was much higher than that in India among the (29.5%) 40+ population.

Overall, there is a wide range of country-level variability in the proportion of the target screening group, and our sample cannot be considered extreme.

### 4.2. Yield of the Screening

The yield of screening for dSVI in the better eye among adults at least 18 years old and among elderly individuals at least 65 years old was estimated to be 3.7% (95% CI: 2.9–4.5; for males 2.3%, 95% CI: 1.3–3.4; for females 4.5% 95% CI: 3.3–5.6) and 9.1% (95% CI: 6.0–12.1; for males 4.3%, 95% CI: 0.9–7.6; for females 12.4% 95% CI: 7.9–17.0), respectively.

The prevalence of dSVI was 5.6%, 3.8%, and 5.4% in Finland among 30+ [40] adults, in the UK among 65+ [38] adults, and in Sweden among 70+ adults [41], respectively. The corresponding values in the target group of our investigation were 4.2%, 9.1%, and 12.1%, respectively. In low-income countries, the prevalence of dSVI was 1.9%, 11.7%, and 1.8%, in Rwanda among 17–39, among 40+ [42], and in India among 15–49 [43], respectively. The corresponding values in the target group of our investigation were 1.4%, 4.8%, and 1.2%, respectively.

The screening yield among adults without glasses or ophthalmologic consultation was higher in our sample than the prevalence of dSVI in Sweden and in the UK. It is highly probable that a significant proportion of dSVI is detectable by population-based screening in Hungary.

According to the representative Hungarian survey among 50+ individuals, the prevalence of dSVI was 14.5% [31]. Because the yield of screening was 6.5% in this age group in our investigation, 45% of the cases are not detected by the usual ophthalmologic care and can be detected by organized screening in the deprived Hungarian population.

### 4.3. Influencing Factors of Screening Positivity

Our results demonstrated the sociodemographic inequality of visual acuity loss. Visual acuity loss of any degree is associated with older age, female sex, and Roma ethnicity. Higher levels of education, employment, and well-treated diabetes were found to be protective factors. The risk factor role of higher age and Roma ethnicity was also observed for the outcome of dSVI. A similar risk factor pattern was observed for elderly individuals over 65 years of age, although the role of employment could not be studied, and the role of education was not significant in this age group.

These observations are in agreement with the known risk factors for visual acuity loss. Surveys in the US, Japan, and Mexico suggested that older age is associated with higher odds of visual impairment [44,45,46,47]. Female sex was an independent risk factor for low vision in studies carried out in Singapore [48,49]. The lower risk among more educated individuals has been convincingly demonstrated by a series of reports [46,50,51]. The protective role of employment was determined in Norway and in South Korea [52,53]. The risk factor role of Roma ethnicity was explored in a former Hungarian study [54]. A high risk among indigenous populations was also detected in Brazil [55]. The protective role of proper care of diabetes mellitus was demonstrated in the United States [56] and Iran [57] as well.

### 4.4. Implications

Because the known prevalence of dSVI is highly variable by country and social strata, screening focusing on people without known dSVI yields a highly variable number of subjects who could benefit from screening by the same dimensions. Therefore, the screening recommendations should be tailored by monitoring for dSVI prevalence or by targeted surveys on the prevalence of dSVI. The lower the prevalence of known dSVI, the more important it would be to organize population-based, active screening.

On the other hand, unknown dSVI seems to be more prevalent among less affordable patients. It seems that people with a lower socioeconomic status could benefit more from the organized approach.

Considering the safe, accurate nature of the screening methods and the fact that the only significant resource need is the organization of population-based screening (calling for and investigating adults), the more screening is implemented into the regular general health check, the most advantageous the cost effectiveness of the screening.

Consequently, not the necessity but the implementation protocol of screening organizations should be the concern in less affordable countries and populations. In a less affordable setting, the affordable countries’ practice, where screening is recommended despite the uncertainties of its benefits [27,58,59], should be followed by integrating visual acuity loss screening into the general health check approach to realize potential health gains. According to this Hungarian study, 45% of dSVI is screening-detectable in the 50+ population, and 59.2% of the screening-detectable cases are from less than 65-year-old adults.

### 4.5. Strengths and Limitations

This investigation was a secondary data analysis on a dataset that was designed to evaluate the relative effectiveness of GPC compared to solo general medical practices (34). A fairly large dataset that provides adequate statistical power was utilized. On the other hand, we faced the obvious limitations of an investigation designed to not evaluate visual acuity loss screening. Namely, there is a lack of data on the pathological conditions behind visual acuity loss among screening-positive subjects and among persons who were not targeted by the screening because of their known ophthalmologic care. Additionally, the sample size of 65+ participants resulted in very wide 95% CIs and the high probability of type 2 errors in statistical inference. Altogether, using this design, we could estimate the yield of a population-based visual acuity loss screening in a setting that differs from the setting from which the recent recommendations originate and determine the high-risk groups where the screening is more effective.

## 5. Conclusions

International recommendations based on Western European and North American studies do not recommend organized screening for visual acuity loss in primary care. According to our study among deprived Hungarian adults, the prevalence of unknown visual acuity loss of at least 0.5 is 3.7% and 9.1% in adults and in the above-65 population, respectively. In this setting, male sex, employment, higher-than-primary education, and properly treated diabetes proved to be protective, while higher age, Roma ethnicity, and untreated hypertension were risk factors. These results demonstrate that adults in Hungary could benefit from regular visual acuity screening.

The high rate of undiagnosed visual acuity loss in this population suggests that primary care should implement population-based screening among adults more rigorously, and the performance of general medical practices should be supported by regular monitoring of this service.

Taking into consideration the huge differences between the health care and the population’s social status of the recommendation-establishing countries and Hungary, which represent non-high-income countries, the discouraging statement of the USPSTF on visual acuity loss screening should not discourage general practitioners from organizing population-based screening for visual acuity loss. Our observation of a higher risk of unknown visual acuity loss among patients with improper care and relative deprivation, even in Hungary, suggests this conclusion.

## Figures and Tables

**Figure 1 healthcare-11-01941-f001:**
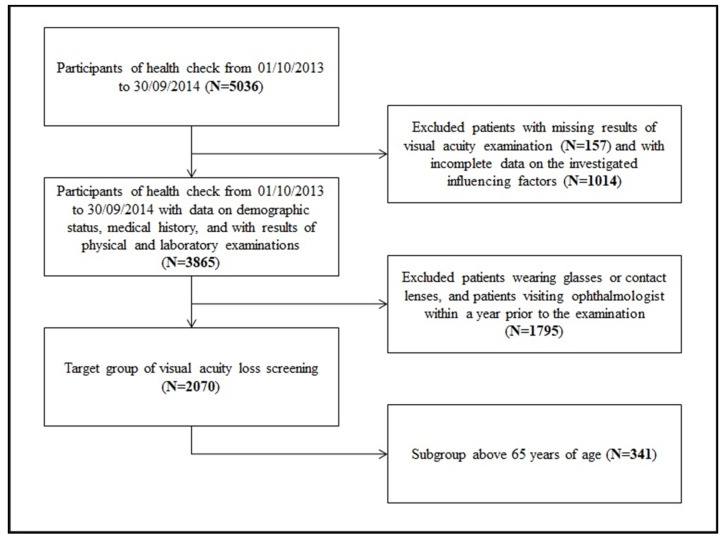
Process of sampling.

**Figure 2 healthcare-11-01941-f002:**
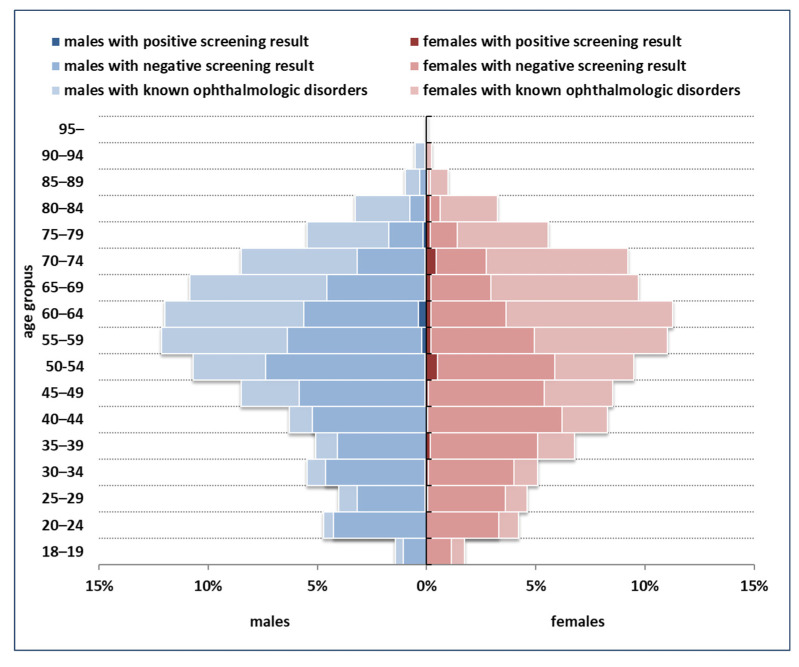
Demographic structure of the population targeted by screening, and the yield of screening across strata.

**Table 1 healthcare-11-01941-t001:** Characteristics of the study population.

		Adults (18+ Years of Age)	Above 65 Years of Age
Sex	Male	768 (37.1%)	140 (41.1%)
	Female	1302 (62.9%)	201 (58.9%)
Age	mean (±SD)	47.8 (±15.9)	71.9 (±5.52)
Education	non-completed primary school	233 (11.3%)	45 (13.2%)
	completed primary school	640 (30.9%)	148 (43.4%)
	secondary school without graduation	511 (24.7%)	57 (16.7%)
	secondary school with graduation	533 (25.7%)	70 (20.5%)
	higher education	153 (7.4%)	21 (6.2%)
Employment	Unemployed	1079 (52.1%)	336 (98.5%)
	Employed	991 (47.9%)	5 (1.5%)
Ethnicity	non-Roma	1825 (88.2%)	334 (97.9%)
	Roma	245 (11.8%)	7 (2.1%)
Hypertension	Normotensive	882 (42.6%)	31 (9.1%)
	untreated hypertension	279 (13.5%)	47 (13.8%)
	insufficiently treated hypertension	487 (23.5%)	144 (42.2%)
	properly treated hypertension	422 (20.4%)	119 (34.9%)
Diabetes	Normoglycemic	1786 (86.3%)	246 (72.1%)
	untreated diabetes	75 (3.6%)	24 (7.0%)
	insufficiently treated diabetes	128 (6.2%)	41 (12.0%)
	properly treated diabetes	81 (3.9%)	30 (8.8%)
Visual acuity	visual acuity ≥ 0.5	1994 (96.3%)	310 (90.9%)
	visual acuity < 0.5	76 (3.7%)	31 (9.1%)
Total number of participants		2070 (100%)	341 (100%)

**Table 2 healthcare-11-01941-t002:** Factors correlated to visual acuity in the entire target group and in the subgroup above 65 years of age by multivariate linear regression models *.

Influencing Factors		Adults (18+ Years of Age)	Above 65 Years of Age
Sex	Male	0 (reference)	0 (reference)
	Female	1.27 [0.35; 2.18]	3.85 [0.5; 7.2]
Age	Year	0.15 [0.12; 0.19]	0.39 [0.1; 0.67]
Education	non-completed primary school	0 (reference)	0 (reference)
	completed primary school	0.45 [−1.03; 1.94]	1.15 [−3.83; 6.12]
	secondary school without graduation	−2.06 [−3.64; −0.47]	−2.58 [−8.76; 3.59]
	secondary school with graduation	−2.08 [−3.65; −0.51]	−1.22 [−6.94; 4.51]
	higher education	−1.82 [−3.89; 0.25]	−0.24 [−7.91; 7.42]
Employment	Unemployed	0 (reference)	---
	Employed	−1.33 [−2.25; −0.4]	nc
Ethnicity	non-Roma	1 (reference)	1 (reference)
	Roma	2.6 [1.22; 3.97]	24.79 [13.83; 35.76]
Hypertension	Normotensive	0 (reference)	0 (reference)
	untreated hypertension	0.31 [−1.08; 1.7]	0.76 [−5.79; 7.31]
	insufficiently treated hypertension	0.36 [−0.92; 1.64]	1.17 [−4.46; 6.8]
	properly treated hypertension	−0.6 [−1.87; 0.68]	0.02 [−5.7; 5.74]
Diabetes mellitus	Normoglycemic	0 (reference)	0 (reference)
	untreated diabetes	0.68 [−1.61; 2.98]	7.3 [1.29; 13.31]
	insufficiently treated diabetes	0.37 [−1.45; 2.19]	1.73 [−3.06; 6.51]
	properly treated diabetes	−2.84 [−5.08; −0.6]	−0.8 [−6.38; 4.78]

* Adjusted linear regression coefficients with corresponding [95% CIs]; nc: not included in the model.

**Table 3 healthcare-11-01941-t003:** Factors associated with visual impairment (with less than 0.5 visual acuity) in the entire target group and in the subgroup above 65 years of age by multivariate logistic regression models.

Influencing Factors		Adults (18+ Years of Age) *	Above 65 Years of Age *
Sex	Male	1 (reference)	1 (reference)
	Female	1.75 [0.98; 3.14]	2.56 [0.91; 7.16]
Age	Year	1.06 [1.04; 1.08]	1.1 [1.03; 1.18]
Education	non-completed primary school	1 (reference)	1 (reference)
	completed primary school	1.14 [0.59; 2.19]	1.16 [0.38; 3.56]
	secondary school without graduation	0.61 [0.25; 1.49]	0.3 [0.03; 3.10]
	secondary school with graduation	0.35 [0.13; 0.93]	0.51 [0.10; 2.44]
	higher education	1.18 [0.41; 3.37]	1.7 [0.33; 8.83]
Employment	Unemployed	1 (reference)	---
	Employed	0.57 [0.30; 1.12]	nc
Ethnicity	non-Roma	1 (reference)	1 (reference)
	Roma	2.80 [1.41; 5.55]	7.81 [1.36; 44.68]
Hypertension	Normotensive	1 (reference)	1 (reference)
	untreated hypertension	0.50 [0.20; 1.26]	0.96 [0.08; 12.3]
	insufficiently treated hypertension	0.80 [0.40; 1.6]	2.56 [0.31; 21.34]
	properly treated hypertension	0.71 [0.35; 1.45]	2.48 [0.29; 21.00]
Diabetes mellitus	Normoglycemic	1 (reference)	1 (reference)
	untreated diabetes	1.16 [0.43; 3.12]	2.03 [0.56; 7.42]
	insufficiently treated diabetes	1.17 [0.52; 2.64]	1.34 [0.39; 4.63]
	properly treated diabetes	0.18 [0.02; 1.32]	0.34 [0.04; 2.87]

* Adjusted odds ratios with corresponding [95% Cis]; nc: not included in the model.

## Data Availability

The dataset analyzed in our investigation is available from the corresponding author on reasonable request. Requests to access this dataset should be directed to janos.sandor@med.unideb.hu.

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
