# Peer review of "Screening for Patients with Visual Acuity Loss in Primary Health Care: A Cross Sectional Study in a Deprived Hungarian Population"

_healthcare, 2023, doi:10.3390/healthcare11131941_

Round 1
Reviewer 1 Report
I consider the work very necessary and draws attention to an important problem.
In the research entitled “ Screening for Patients with Visual Acuity Loss in Primary Health Care: A Cross Sectional Study in A Deprived Hungarian Population” the main question addressed by the research was to evaluate the effectiveness of primary health care-based screening. A study based on the purpose of evaluating the use of screening at the standard level, as it is based on screening for recognition of visual acuity loss, is not applicable, due to specific UN-based criteria based on observations from affordable countries. The studies aimed to evaluate the effectiveness of primary health care-based screening.
In my opinion the topic of the research is original and relevant in the field of screening.
A cross-sectional investigation was carried out among adults who did not wear glasses and did not visit ophthalmologist in a year (N = 2070). Risk factor role of sociodemographic factors and cardiometabolic status for hidden visual acuity loss (VAL) was determined by multivariable linear regression models.
The prevalence of unknown VAL of at least 0.5 was 3.7% and 9.1% in adults and in the above-65 population. Above 65 years, female sex (b = 3.85, 0.50;7.20), age (b = 0.39, 0.10; 0.67), Roma ethnicity (b = 24.79, 13.83; 35.76) and untreated diabetes (b = 7.30, 1.29; 13.31) were associated with VAL.
Considering the huge differences between health care and population social status of the recommendation establishing countries and Hungary represented nonhigh-income countries, the uncertain recommendation on VAL screening cannot discourage general practitioners from organizing population-based screening for VAL.
Conclusions based on the question posed are consistent with the evidence and arguments presented.
The references are appropriate, I have no comments to the tables and figures, they are clear and transparent.
Author Response
Dear Reviewer,
Thank you very much for the careful review of our manuscript. Please find enclosed the revised version of the manuscript “Screening for patients with Visual Acuity loss in primary health care: a cross sectional study in a deprived Hungarian population” by Rahul Naresh Wasnik, et al.
Each comment and suggestion has been considered. The corresponding changes and refinements made in the revised paper are summarized in our response after considering each of your suggestion. Answers along with the modifications we made are summarized below (comments/questions of Yours are in capitals).
Sincerely yours, Janos Sandor (on behalf of the authors)
Answers/reflections to the comments of Reviewer-1:
1.
I CONSIDER THE WORK VERY NECESSARY AND DRAWS ATTENTION TO AN IMPORTANT PROBLEM.
Thank you for this comment!
2.
IN THE RESEARCH ENTITLED “SCREENING FOR PATIENTS WITH VISUAL ACUITY LOSS IN PRIMARY HEALTH CARE: A CROSS SECTIONAL STUDY IN A DEPRIVED HUNGARIAN POPULATION” THE MAIN QUESTION ADDRESSED BY THE RESEARCH WAS TO EVALUATE THE EFFECTIVENESS OF PRIMARY HEALTH CARE-BASED SCREENING. A STUDY BASED ON THE PURPOSE OF EVALUATING THE USE OF SCREENING AT THE STANDARD LEVEL, AS IT IS BASED ON SCREENING FOR RECOGNITION OF VISUAL ACUITY LOSS, IS NOT APPLICABLE, DUE TO SPECIFIC UN-BASED CRITERIA BASED ON OBSERVATIONS FROM AFFORDABLE COUNTRIES. THE STUDIES AIMED TO EVALUATE THE EFFECTIVENESS OF PRIMARY HEALTH CARE-BASED SCREENING.
Thank you for this summary!
3.
IN MY OPINION, THE TOPIC OF THE RESEARCH IS ORIGINAL AND RELEVANT IN THE FIELD OF SCREENING.
Thank you for this comment!
4.
A CROSS-SECTIONAL INVESTIGATION WAS CARRIED OUT AMONG ADULTS WHO DID NOT WEAR GLASSES AND DID NOT VISIT OPHTHALMOLOGIST IN A YEAR (N = 2070). RISK FACTOR ROLE OF SOCIODEMOGRAPHIC FACTORS AND CARDIOMETABOLIC STATUS FOR HIDDEN VISUAL ACUITY LOSS (VAL) WAS DETERMINED BY MULTIVARIABLE LINEAR REGRESSION MODELS.
Thank you for this summary!
5.
THE PREVALENCE OF UNKNOWN VAL OF AT LEAST 0.5 WAS 3.7% AND 9.1% IN ADULTS AND IN THE ABOVE-65 POPULATION. ABOVE 65 YEARS, FEMALE SEX (B = 3.85, 0.50;7.20), AGE (B = 0.39, 0.10; 0.67), ROMA ETHNICITY (B = 24.79, 13.83; 35.76) AND UNTREATED DIABETES (B = 7.30, 1.29; 13.31) WERE ASSOCIATED WITH VAL.
Thank you for this summary!
6.
CONSIDERING THE HUGE DIFFERENCES BETWEEN HEALTH CARE AND POPULATION SOCIAL STATUS OF THE RECOMMENDATION ESTABLISHING COUNTRIES AND HUNGARY REPRESENTED NONHIGH-INCOME COUNTRIES, THE UNCERTAIN RECOMMENDATION ON VAL SCREENING CANNOT DISCOURAGE GENERAL PRACTITIONERS FROM ORGANIZING POPULATION-BASED SCREENING FOR VAL.
Thank you for this summary!
7.
CONCLUSIONS BASED ON THE QUESTION POSED ARE CONSISTENT WITH THE EVIDENCE AND ARGUMENTS PRESENTED.
Thank you for this comment!
8.
THE REFERENCES ARE APPROPRIATE, I HAVE NO COMMENTS TO THE TABLES AND FIGURES, THEY ARE CLEAR AND TRANSPARENT.
Thank you for this comment!

Reviewer 2 Report
The present review paper showcases an extensive amount of carefully reviewed bibliographic material, accompanied by a meticulous search effort concerning the topic analyzed. Far from minor issues that require revision, the language used is correct, and the structure is easily comprehensible. I extend my congratulations to the authors for their commendable work. Moreover, I would like to humbly offer a few comments.
General comment (even in the abstract): the notation for the reported results should be accurate in contact with the entire manuscript. The notation should be always the same “OR = value, 95%CI: value; value”, sometimes the “95%” is reported, sometimes not, sometimes “CI” is reported, sometimes don’t… and in the same sentence. Please, be careful with this kind of detail.
Lines 36-38: while it is right, this sentence requires a reference.
Line 81: “Visual acuity loss” is used here, BUT the acronym “VAL” is generated. Please, be constant in the terminology employed in all the manuscripts.
Line 95: similar “age-related macular degeneration” should be AMD previously used on line 38.
Line 143: brand and manufacturer should be reported.
Far from minor issues that require revision, the language used is correct, and the structure is easily comprehensible.
Author Response
Dear Reviewer,
Thank you very much for the careful review of our manuscript. Please find enclosed the revised version of the manuscript “Screening for patients with Visual Acuity loss in primary health care: a cross sectional study in a deprived Hungarian population” by Rahul Naresh Wasnik, et al.
Each comment and suggestion has been considered. The corresponding changes and refinements made in the revised paper are summarized in our response after considering each of your suggestion. Answers along with the modifications we made are summarized below (comments/questions of Yours are in capitals).
Sincerely yours, Janos Sandor (on behalf of the authors)
Answers/reflections to the comments of Reviewer-2:
1.
THE PRESENT REVIEW PAPER SHOWCASES AN EXTENSIVE AMOUNT OF CAREFULLY REVIEWED BIBLIOGRAPHIC MATERIAL, ACCOMPANIED BY A METICULOUS SEARCH EFFORT CONCERNING THE TOPIC ANALYZED. FAR FROM MINOR ISSUES THAT REQUIRE REVISION, THE LANGUAGE USED IS CORRECT, AND THE STRUCTURE IS EASILY COMPREHENSIBLE. I EXTEND MY CONGRATULATIONS TO THE AUTHORS FOR THEIR COMMENDABLE WORK. MOREOVER, I WOULD LIKE TO HUMBLY OFFER A FEW COMMENTS.
Thank you for this comment!
2.
GENERAL COMMENT (EVEN IN THE ABSTRACT): THE NOTATION FOR THE REPORTED RESULTS SHOULD BE ACCURATE IN CONTACT WITH THE ENTIRE MANUSCRIPT. THE NOTATION SHOULD BE ALWAYS THE SAME “OR = VALUE, 95%CI: VALUE; VALUE”, SOMETIMES THE “95%” IS REPORTED, SOMETIMES NOT, SOMETIMES “CI” IS REPORTED, SOMETIMES DON’T… AND IN THE SAME SENTENCE. PLEASE, BE CAREFUL WITH THIS KIND OF DETAIL.
Thank you for this comment! The reporting formula of (OR = value, 95% CI: value; value) has been applied throughout the manuscript. Corrections can be checked in the manuscript revised tracked changes.docx file uploaded to the Journal’s website.
3.
LINES 36-38: WHILE IT IS RIGHT, THIS SENTENCE REQUIRES A REFERENCE.
The first three sentences of the manuscript are related to the reference 1. This reference is inserted at the end of the third sentence. To emphasize it, a paragraph break was inserted after the reference 1.
4.
LINE 81: “VISUAL ACUITY LOSS” IS USED HERE, BUT THE ACRONYM “VAL” IS GENERATED. PLEASE, BE CONSTANT IN THE TERMINOLOGY EMPLOYED IN ALL THE MANUSCRIPTS.
We used the “visual acuity loss” as general term for any form of the pathological status. As it is written in the Methods section, we specified the statistical measure of the visual acuity loss “as continuous outcome variable as visual acuity loss” and for that measure we applied VAL abbreviation. To avoid the misleading writing, the VAL abbreviation has been replaced with cVAL, and using the same approach for dichotomized visual acuity measure, the SVI has been replaced with dSVI, throughout the manuscript. Corrections can be checked in the manuscript revised tracked changes.docx file uploaded to the Journal’s website.
5.
LINE 95: SIMILAR “AGE-RELATED MACULAR DEGENERATION” SHOULD BE AMD PREVIOUSLY USED ON LINE 38.
The “age-related macular degeneration” has been replaced with AMD abbreviation.
6.
LINE 143: BRAND AND MANUFACTURER SHOULD BE REPORTED.
The “Kettesy-type table” is the type of a diagnostic equipment. Many manufacturers support the GPs. There is no difference between the Kettesy-tables produced by different manufacturers. The sentence has not been corrected. We hope that You can accept it!
7.
COMMENTS ON THE QUALITY OF ENGLISH LANGUAGE: FAR FROM MINOR ISSUES THAT REQUIRE REVISION, THE LANGUAGE USED IS CORRECT, AND THE STRUCTURE IS EASILY COMPREHENSIBLE.
The manuscript was edited for proper English language, grammar, punctuation, spelling, and overall style by one or more of the highly qualified native English speaking editors at American Journal Experts. The certificate has been uploaded to the website of the Healthcare journal.

Reviewer 3 Report
Line 57-58: “Considering the very high prevalence of undiagnosed visual impairment and the safety and low cost of visual acuity checking…” The sentence is not clear. Please revise.
Line 153-155: Please clarify how to define a “loss” of visual acuity in VAL calculation. What was the criterion of “health” or “without visual loss”? And Please clarify the definition of SVI. What was “0.5” visual acuity? LogMAR or Snellen?
Results: P values shall be presented in the Results section. Furthermore, I will suggest to include VAL as a binary outcome to perform a logistic regression model to calculate the OR.
Table 2: Please clarify the reasons why the factors were included into multivariate analysis.
Table 2 and 3: The sample size of participants with Roma ethnicity was too small (n=7), particularly in the “Above 65 years of age” group. There may not be enough power to draw a conclusion that “Roma ethnicity is a risk factor of VAL or SVI”.
Minor improvement is reqiured.
Author Response
Dear Reviewer,
Thank you very much for the careful review of our manuscript. Please find enclosed the revised version of the manuscript “Screening for patients with Visual Acuity loss in primary health care: a cross sectional study in a deprived Hungarian population” by Rahul Naresh Wasnik, et al.
Each comment and suggestion has been considered. The corresponding changes and refinements made in the revised paper are summarized in our response after considering each of your suggestion. Answers along with the modifications we made are summarized below (comments/questions of Yours are in capitals).
Sincerely yours, Janos Sandor (on behalf of the authors)
Answers/reflections to the comments of Reviewer-3:
1.
LINE 57-58: “CONSIDERING THE VERY HIGH PREVALENCE OF UNDIAGNOSED VISUAL IMPAIRMENT AND THE SAFETY AND LOW COST OF VISUAL ACUITY CHECKING…” THE SENTENCE IS NOT CLEAR. PLEASE REVISE.
The sentence has been rewritten. The original sentence “Considering the very high prevalence of undiagnosed visual impairment and the safety and low cost of visual acuity checking, population-based screening seems to be an important tool to reduce the consequences of undiagnosed visual impairment [16].” has been replaced with “Considering that the prevalence of undiagnosed visual impairment is very high, and the visual acuity checking is safe and cheap, population-based screening seems to be an important tool to reduce the consequences of undiagnosed visual impairment [16].”.
2.
LINE 153-155: PLEASE CLARIFY HOW TO DEFINE A “LOSS” OF VISUAL ACUITY IN VAL CALCULATION. WHAT WAS THE CRITERION OF “HEALTH” OR “WITHOUT VISUAL LOSS”? AND PLEASE CLARIFY THE DEFINITION OF SVI. WHAT WAS “0.5” VISUAL ACUITY? LOGMAR OR SNELLEN?
Thank you very much for this comment!
Visual acuity was quantified by Kettesy chart which is a Snellen chart, according to the standard ophthalmologic practice. The result from this examination (visual acuity) was used to calculate the visual acuity loss (VAL=1-VA) as a continuous parameter without specifying healthy and unhealthy subjects. For logistic regression modelling, VAL was dichotomized using the 0.5 vision as threshold.
The sentences of 2.3 section have been corrected:
Original sentences:
“Visual acuity was measured with a Snellen-type Kettesy table consisting of optotypes. (Each row of these test charts is designed from larger optotypes into smaller optotypes to evaluate patients reading and observing abilities.) Visual acuity was expressed as the ratio of the distance from the patient to the test chart and the distance at which the smallest optotype that the patient can read can be detected for a healthy eye.”
and
“All participants were characterized with a continuous outcome variable as visual acuity loss (VAL, assessing the severity of total ocular damage) and with a dichoto-mous outcome variable as severe (visual acuity is less than 0.5) visual impairment (SVI).”
Corrected sentences:
“Visual acuity was measured with a Kettesy-type table consisting of optotypes ac-cording to the standard ophthalmologic practice. (Each row of these test charts is de-signed from larger optotypes into smaller optotypes to evaluate patients reading and observing abilities.) Visual acuity (VA) was expressed as the ratio of the distance from the patient to the test chart and the distance at which the smallest optotype that the patient can read can be detected for a healthy eye.”
and
“All participants were characterized with a continuous outcome variable as visual acuity loss (cVAL, assessing the severity of total ocular damage by calculating 1-VA) and with a dichotomous outcome variable as severe (cVAL<visual acuity is less than 0.5) visual impairment (dSVI).”
3.
RESULTS: P VALUES SHALL BE PRESENTED IN THE RESULTS SECTION.
The statistical inference was drawn by 95% confidence intervals and not by p-values, because our intention was to report the precision of estimates (for linear regression coefficients and for odds ratios). Mainly, because of the small sample size among 65+ participants. Reporting only, the higher than 0.05 p-values could not demonstrate the precision of the estimates.
4.
FURTHERMORE, I WILL SUGGEST TO INCLUDE VAL AS A BINARY OUTCOME TO PERFORM A LOGISTIC REGRESSION MODEL TO CALCULATE THE OR.
The SVI is the dichotomized version of the VAL. To avoid the misleading definitions, the sentence in the Methods 2.3 Visual acuity assessment has been corrected.
Original sentence:
“All participants were characterized with a continuous outcome variable as visual acuity loss (VAL, assessing the severity of total ocular damage) and with a dichotomous outcome variable as severe (visual acuity is less than 0.5) visual impairment (SVI).”
Corrected sentence:
“All participants were characterized with a continuous outcome variable as visual acuity loss (cVAL, assessing the severity of total ocular damage) and with a dichoto-mous outcome variable as severe (cVAL<0.5) visual impairment (dSVI).”
5.
TABLE 2: PLEASE CLARIFY THE REASONS WHY THE FACTORS WERE INCLUDED INTO MULTIVARIATE ANALYSIS.
The knowledge on the importance of socio-demographic status is summarized in the first two paragraphs of the introduction.
The regular ophthalmologic checking as a part of hypertension and diabetes mellitus care is a confounding factor for our analysis. Because these disorders have very high prevalence, these factors had to be involved in the multivariable modelling. The explicit statement for this argumentation, the relevant sentence in 2.2 sections has been modified.
Original sentence:
“Patients’ blood pressure and fasting blood glucose concentration were measured, and they were categorized into one of the following subgroups”
Corrected sentence:
“Because the regular ophthalmologic examination is a compulsory part of the care for high prevalence chronic diseases (hypertension and diabetes mellitus), patients’ blood pressure and fasting blood glucose concentration were measured, and the cardiomet-abolic history has been registered in the GHC, and they were categorized into one of the following subgroups”
6.
TABLE 2 AND 3: THE SAMPLE SIZE OF PARTICIPANTS WITH ROMA ETHNICITY WAS TOO SMALL (N=7), PARTICULARLY IN THE “ABOVE 65 YEARS OF AGE” GROUP. THERE MAY NOT BE ENOUGH POWER TO DRAW A CONCLUSION THAT “ROMA ETHNICITY IS A RISK FACTOR OF VAL OR SVI”.
A sentence has been inserted into the “Strengths and limitations” section. The small statistical power has been acknowledged. The inserted sentence: “Additionally, the sample size of 65+ participants resulted in very wide 95% CIs, and the high probability of type 2 errors in statistical inference.”
7.
COMMENTS ON THE QUALITY OF ENGLISH LANGUAGE: MINOR IMPROVEMENT IS REQUIRED.
The manuscript was edited for proper English language, grammar, punctuation, spelling, and overall style by one or more of the highly qualified native English speaking editors at American Journal Experts. The certificate has been uploaded to the website of the Healthcare journal.

Reviewer 4 Report
I have read this manuscript with interest. The authors tried to evaluate the effectiveness of primary health care-based screening in A Deprived Hungarian Population. On the whole, this article has some novelty and scientific value, but there are still some problems, so I suggest the authors revise the article.
I have the following suggestions:
â‘ Language: On the whole, I can understand what the author is trying to express in this manuscript, but there are some places where the expression is not smooth.
â‘¡The clarity of image 1 needs to be improved
â‘¢I think the content of the method part is not detailed enough. I hope the authors can explain their research steps in more detail, so that other researchers can easily replicate this study or similar studies.
â‘£In Table 3, Authors mentioned "insufficiently treated hypertension" and "properly treated hypertension". They should provide a specific definition of "insufficiently treated hypertension" and "properly treated hypertension" instead, and other similar parts of the text should also do the same.
Minor editing of English language required
Author Response
Dear Reviewer,
Thank you very much for the careful review of our manuscript. Please find enclosed the revised version of the manuscript “Screening for patients with Visual Acuity loss in primary health care: a cross sectional study in a deprived Hungarian population” by Rahul Naresh Wasnik, et al.
Each comment and suggestion has been considered. The corresponding changes and refinements made in the revised paper are summarized in our response after considering each of your suggestion. Answers along with the modifications we made are summarized below (comments/questions of Yours are in capitals).
Sincerely yours, Janos Sandor (on behalf of the authors)
Answers/reflections to the comments of Reviewer-4:
1.
I HAVE READ THIS MANUSCRIPT WITH INTEREST. THE AUTHORS TRIED TO EVALUATE THE EFFECTIVENESS OF PRIMARY HEALTH CARE-BASED SCREENING IN A DEPRIVED HUNGARIAN POPULATION. ON THE WHOLE, THIS ARTICLE HAS SOME NOVELTY AND SCIENTIFIC VALUE, BUT THERE ARE STILL SOME PROBLEMS, SO I SUGGEST THE AUTHORS REVISE THE ARTICLE.
Thank you for this comment!
I have the following suggestions:
2.
LANGUAGE: ON THE WHOLE, I CAN UNDERSTAND WHAT THE AUTHOR IS TRYING TO EXPRESS IN THIS MANUSCRIPT, BUT THERE ARE SOME PLACES WHERE THE EXPRESSION IS NOT SMOOTH.
The manuscript was edited for proper English language, grammar, punctuation, spelling, and overall style by one or more of the highly qualified native English speaking editors at American Journal Experts. The certificate has been uploaded to the website of the Healthcare journal.
3.
THE CLARITY OF IMAGE 1 NEEDS TO BE IMPROVED
The term “Target group of visual acuity examination” has been replaced with “Target group of visual acuity loss screening”.
4.
I THINK THE CONTENT OF THE METHOD PART IS NOT DETAILED ENOUGH. I HOPE THE AUTHORS CAN EXPLAIN THEIR RESEARCH STEPS IN MORE DETAIL, SO THAT OTHER RESEARCHERS CAN EASILY REPLICATE THIS STUDY OR SIMILAR STUDIES. IN TABLE 3, AUTHORS MENTIONED "INSUFFICIENTLY TREATED HYPERTENSION" AND "PROPERLY TREATED HYPERTENSION". THEY SHOULD PROVIDE A SPECIFIC DEFINITION OF "INSUFFICIENTLY TREATED HYPERTENSION" AND "PROPERLY TREATED HYPERTENSION" INSTEAD, AND OTHER SIMILAR PARTS OF THE TEXT SHOULD ALSO DO THE SAME.
The explanation for categories has been modified.
Original sentence:
“Patients’ blood pressure and fasting blood glucose concentration were measured, and they were categorized into one of the following subgroups: no existing disease or treatment (normotensive, normoglycemic), diagnosed and properly treated, diagnosed but inadequately treated, or unknown and therefore untreated disease group.”
Corrected sentence:
“Patients’ blood pressure and fasting blood glucose concentration were measured, and the cardiometabolic history has been registered in the GHC, and they were categorized into one of the following subgroups: (a) no existing disease or treatment (normoten-sive, normoglycemic), (b) diagnosed and properly treated, (c) diagnosed but inade-quately treated, or (d) unknown and therefore untreated disease group.”
5.
COMMENTS ON THE QUALITY OF ENGLISH LANGUAGE: MINOR EDITING OF ENGLISH LANGUAGE REQUIRED.
The manuscript was edited for proper English language, grammar, punctuation, spelling, and overall style by one or more of the highly qualified native English speaking editors at American Journal Experts. The certificate has been uploaded to the website of the Healthcare journal.

Round 2
Reviewer 3 Report
I have no further comment.